# Comparison of Type I and Type III Collagen Concentration between *Oreochromis mossambicus* and *Oreochromis niloticus* in Relation to Skin Scaffolding

**DOI:** 10.3390/medicina59061002

**Published:** 2023-05-23

**Authors:** Bogdan Ciornei, Adrian Vaduva, Vlad Laurentiu David, Diana Popescu, Dan Dumitru Vulcanescu, Ovidiu Adam, Cecilia Roberta Avram, Alina Cornelia Pacurari, Eugen Sorin Boia

**Affiliations:** 1Department of Pediatric Surgery and Orthopedics, “Victor Babes” University of Medicine and Pharmacy, 300002 Timisoara, Romania; bogdan.ciornei@umft.ro (B.C.); david.vlad@umft.ro (V.L.D.); adamovidiu29@yahoo.com (O.A.); boiaeugen@yahoo.com (E.S.B.); 2Department of Pathology, Methodological Research Center ANAPATMOL, “Victor Babes” University of Medicine and Pharmacy, 300002 Timisoara, Romania; 3Department of Pediatric Surgery, “Louis Turcanu” Emergency Children’s Hospital, 300011 Timisoara, Romania; popescudia90@gmail.com; 4Multidisciplinary Research Center on Antimicrobial Resistance (Multi-Rez), “Victor Babes“ University of Medicine and Pharmacy, 300002 Timisoara, Romania; dan.vulcanescu@umft.ro; 5Department of Residential Training and Post-University Courses, “Vasile Goldis” Western University, 300002 Arad, Romania; avram.cecilia@uvvg.ro; 6Department of Internal Medicine, Medlife Hyperclinic, 300551 Timisoara, Romania

**Keywords:** skin scaffold, grafting, burn treatment, collagen content, *Tilapia mossambica*, *Tilapia niloticus*

## Abstract

*Background and Objectives:* Skin scaffolding can be done using allografts and autografts. As a biological allograft, the skin of *Oreochromis niloticus* (ON) has been used due to its high type I and III collagen content. *Oreochromis mossambicus* (OM) is also a member of the Oreochromis family, but not much is known regarding its collagen content. As such, this study aimed to assess and compare the collagen content of the two fish species. *Materials and Methods:* This is a crossover study comparing the skin collagen contents of the two fish. Young fish were chosen, as they tend to have higher collagen concentrations. The skin samples were sterilized in chlorhexidine and increasing glycerol solutions and analyzed histochemically with Sirius red picrate under polarized light microscopy. *Results*: 6 young ON and 4 OM specimens were used. Baseline type I collagen was higher for OM, but at maximum sterilization it was higher for ON, with no differences in between Type III collagen was higher for OM across all comparisons with the exception of the last stage of sterilization. Generally, collagen concentrations were higher in highly sterilized samples. *Conclusions*: OM skin harvested from young fish, with its greater collagen III content may be a better candidate for use as a biological skin scaffold in the treatment of burn wounds, compared to ON.

## 1. Introduction

Burns represents one of the leading traumatic injuries in the world. In the US, around 450,000 people suffer from burns every year, while around the world approximately 195,000 deaths are accountable to these injuries [1,2]. In adults, workplace accidents are the leading cause of burns, while in children neglect and abuse are the leading causes. For pediatric patients, scald injuries are the most frequent, approximately 100,000 seeking medical care due to this incident every year in the US [3].

Burn covering materials are essential for the healing process of these wounds. They can either be synthetic or biological. The most used biologic scaffolds are represented by allografts and autografts, but their supply and use are limited by donor incompatibility, in the case of allografts, or by morbidity from additional trauma of the skin through autografts [4,5,6]. 

Fish skin and its components have long been proposed as temporary grafts for burn wounds. Acellular fish skin has been shown to promote faster healing rates in comparison to bovine skin grafts, with reduced contraction rates, faster epithelialization, and better integration into the wound bed [7,8]. Electrospun fish collagen combined with inorganic components such as Cobalt or Copper ions could prove to enhance neo-vascularization in the first stages of wound healing [9].

The Nile Tilapia (*Oreochromis niloticus*) (ON) skin has been proposed as a possible biologic scaffold to be used in burns for several years [10,11,12,13]. Its use in burn healing has been evaluated to some extent, but there is a long way till it will gain universal acceptance. Its extracellular matrix (ECM) is comprised of type I and III collagen, similar to human ECM, rendering it a cheap, readily available, and possibly a biocompatible scaffold for wounds [14,15,16,17,18]. 

*Tilapia Mossambica* (*Oreochromis mossambicus*) (OM) is also a subspecies of the Oreochromis family, with a natural habitat located in the same regions of Central and East Africa. In other parts of the world, it is considered, like the Nile Tilapia, an invasive species with adverse effects on local ecosystems through competition for food [19,20,21]. For these reasons, the sources for both fish are limited in most of the regions outside their natural habitat, and alternatives to ON skin, like OM skin, may be useful. Not much is known in the literature about the collagen content of this fish’s skin, the Nile Tilapia being the more studied specimen.

The purpose of this study was to compare the two Tilapia subspecies, OM and ON in regards to collagen type I and III content, before and after sterilization using increasing chlorhexidine and concentrations of glycerol solutions. Additionally, we have investigated the differences between collagen type I and collagen type III ratios in both species. We conducted this study with the ultimate goal of discovering an alternative source for a biological scaffold of fish origin.

## 2. Materials and Methods

### 2.1. Preparation and Sterilization

We performed a crossover study, in which we used 10 young Tilapia fish (6 ON, 4 OM), obtaining one skin specimen from each fish. The specimens were harvested from the lateral side of each fish, resulting in 10 total pieces of skin. We have done this to be able to trace the origin of the specimens and to observe the direct histologic effect of the multiple stages of the sterilization process.

The sterilization protocol is performed after the one described by Alves et al. with minor modifications [16] and the steps we took can be seen in Table 1.

After completion of each of these steps, a 2 × 1 cm sample of tissue was harvested from each specimen to undergo histochemical analysis.

### 2.2. Microbiological Analysis

For the microbiological analysis, a 0.5 × 0.5 cm fragment was taken from each sample after completion of each stage of sterilization and imprinted consecutively on Petri dishes containing Blood-agar, Hicrome UTI agar, Chapman medium, and MacConkey agar, incubated at 36.3 °C in a Memmert incubator (Memmert GmbH & Co., Ltd. KG, Aussere Rittersbacher Strasse 38 D-91126 Schwabach, Germany). The dishes were evaluated for bacterial growth after 24, 48, and 72 h [22].

### 2.3. Histochemical Analysis

The skin samples were fixed in formalin and further embedded in paraffin as per the usual technique.

Histochemical analysis was performed using Sirius Red picrate staining (Bio-Optica Milano S.p.A. Via San Faustino 58-20134 Milano, Italy) in the evaluation process of type I and III collagen components under polarized light. The staining was performed according to the manufacturer’s instructions. Five photomicrographs were taken from each sample at ×20 magnification, belonging to each sterilization stage, adding up to 200 photomicrographs that were submitted for analysis. Polarized light microscopy images were obtained on a Leica DMD108 microscope (Leica Microsystems, Ernst-Leitz-Strasse 17-35, 35578 Wetzlar, Germany) and saved as uncompressed TIFF images.

These images were then run through an automated image analysis protocol in Icy Bioimage Analysis software [19]. Briefly, each image was split into two additional images representing the red and green channels (Figure 1).

Automatic segmentation of the positive polarized areas was performed afterward, resulting in regions of interest (ROI) for which statistics were saved to a xls file for statistical analysis.

The data obtained by the Icy BioImage Analysis Software included the total area (µm^2^) covered in the ROI on the respective channel (type I or type III collagen). After determining the total surface covered by the two collagen types, we calculated a ratio between collagen type I and III, on each sample [23].

### 2.4. Statistical Analysis

Statistical analysis was performed using SPSS IBM Statistics (v26) (IBM Corp. Released 2019. IBM SPSS Statistics for Windows, Version 26.0. Armonk, NY, USA: IBM Corp). 

Sample size was calculated using the G*Power software (v 3.1.9.6), according to the ARRIVE Guidelines, using an a priori test to calculate the minimum sample size for a high effect size (0.5) and a power of 80%. The result indicated that a minimum of 8 samples were needed. As such, 10 fish were chosen.

The raw data was formatted in a wide format and tested for normality and sphericity. Repeated measures ANOVA with Bonferroni correction was used to assess in-group differences, with the default pairwise *t*-test used for multiple comparisons. A Student’s *t*-test assuming unequal variances was used at each stage to compare the two fish. The significance level was set to 95%, (*p* values lower than 0.05).

The xls file computed by the IcyBioImage software provided multiple determinations of areas covered by the red and green channels. We then calculated the mean area of collagen on the respective channel for each sample, by dividing the total area covered (µm^2^) by the number of individual surfaces. In this way, we calculated the mean area covered by collagen (type I and type III) in each sample and used it to evaluate the differences between the two species. Also with this new data, we have calculated the ratio between the two collagen types.

### 2.5. Ethical Considerations

The study protocol was presented and approved by the Committee on Research Ethics of the University of Medicine and Pharmacy “Victor Babes” Timisoara, under number 61/30.08.2021, being in line with the European directive 2010/63/EU European Union OJ L276/33 and The Helsinki Declaration.

## 3. Results

We used 10 young fish, out of which 6 were in the ON group, with weights ranging from 92–160 g, and 4 in the OM group, weighing between 129–340 g.

### 3.1. Microbiological Analysis

Among the 10 fish specimens subjected to microbiological analysis, contamination was observed in two samples (1 ON and 1 OM) due to mishandling, and thus, were not further examined. The remaining 8 samples were inoculated onto the previously mentioned 4 culture media. Notably, none of the 32 samples revealed any signs of bacterial growth on the Petri dishes containing Blood-agar, Hicrome UTI agar, Chapman medium, and MacConkey agar, irrespective of the incubation period (24, 48, and 72 h).

### 3.2. Collagen Type I Histochemical Analysis

Using Student’s *t* test to compare the collagen type I evolution between the two species showed that there is a statistically significant difference between ON and OM at baseline (natur, *p* = 0.017, ON < OM) and at the last stage of sterilization (99% glycerol, *p* = 0.022, OM < ON), as seen in Table 2 and Figure 2.

Table 3 contains information and comparisons regarding in-group sterilization, assessed by the repeated measures ANOVA. For the ON type, natur type I collagen content was higher the higher with sterilization stage (*p* < 0.001), with the exception of 50% glycerol vs. 75% glycerol samples, which showed no statistical difference (*p* = 0.394). For the OM fish, the following differences were observed: natur vs. 99% glycerol (*p* = 0.001) and 50% glycerol vs. 99% glycerol (*p* = 0.028), with collagen concentration being higher in the more sterilized sample.

### 3.3. Collagen Type III Histochemical Analysis

The outcomes of the *t*-test indicate a notable variation in the mean collagen type III content between the two species within the baseline group, with OM exhibiting over 40% higher concentration in contrast to ON (natur, *p* < 0.001). Additionally, Table 4 and Figure 3 reveal that collagen III levels were significantly elevated at the initial stage of sterilization (50% glycerol, *p* = 0.026) and second stage (75% glycerol, *p* = 0.16). Nonetheless, this difference was not statistically significant in the 99% Glycerol group.

Table 5 contains information and comparisons regarding in-group sterilization using the repeated measures ANOVA. For ON, a difference was observed between natur concentrations and each sterilization stage (*p* = 0.001 or less), with the exception of 50% glycerol vs. 75% glycerol (*p* = 0.065) and 75% glycerol vs. 99% glycerol (*p* = 0.427), with collagen concentration being higher in the more sterilized samples. For OM, the following differences were observed: natur vs. 99% glycerol (*p* = 0.005), with collagen concentration being higher in the more sterilized sample, 50% glycerol vs. 99% glycerol (*p* = 0.037) and 75% glycerol vs. 99% glycerol (*p* < 0.001), with collagen concentration being higher in the less sterilized samples.

### 3.4. Collagen I/III Ratio

As expressed in Table 6 and Figure 4, the collagen ratio is significantly different between the two species of fish at the *t* test. OM presented a lower ratio for natur and 50% glycerol (*p* < 0.001), yet the differences could not be considered statistically significant in regards to 75% glycerol (*p* = 0.050) and 99% glycerol (*p* = 0.244). This is due to higher type III collagen concentrations.

Table 7 contains information and comparisons regarding in-group sterilization using the repeated measures ANOVA. For ON, significant differences were observed, with higher ratios being observed in more sterilized samples (*p* = 0.006 or less), with the exception of natur vs. 75% glycerol (*p* = 0.480). For the OM samples, the following statistical differences were observed: natur vs. 99% glycerol (*p* < 0.001), 50% glycerol vs. 99% glycerol (*p* < 0.001) and 75% glycerol vs. 99% glycerol (*p* < 0.001).

## 4. Discussion

Information about the collagen content of OM skin is scarce. Our inquiry into the literature has produced very little insight into the extent of OM as a subject of study. Liu et al., have studied the collagen content of OM bones, but not the skin [24]. Most articles have relied on the use of ON skin to study possible healing effects of the collagen contained within, but compared to our study, the determination of the amount of collagen was done by purification extraction methods such as enzymatic reactions, hot water, acetic-acid methods, sodium hydroxide methods, and others [25,26,27]. All of these studies have used mature fish, and it seems that after performing the above-mentioned processes of extraction, they have produced only collagen with a biochemical composition consistent with type I. However, for regeneration of the skin to happen, a biological scaffold needs type III collagen as well [28].

Other wound dressings that help skin regeneration after burns are represented by biosynthetic coverings, which have the role of temporarily or permanently closing the wound in order for it to heal. Some use the patient’s own cells to favor this process, others are comprised of an acellular collagen matrix which offers support for fibroblast and endothelial cell migration and proliferation [29]. Key characteristics of wound coverings are the ability to prevent heat and water loss and at the same time to permit exudate removal. These products also need to be biocompatible, have good adherence to the wound bed, and be free of microbial or viral colonization [30].

The biocompatibility of Tilapia fish collagen has been studied by several researchers [13,31,32,33], through in vitro studies, reaching the conclusion that indeed collagen derived from Tilapia fish skin or scales is compatible with human cells, producing no severe immunogenic reaction. Also, it seems that in vitro, Tilapia collagen might have an antimicrobial effect as well [34].

Previous studies involving the Nile Tilapia have shown the potential of this biological scaffold for the treatment of burns. It has the main elements that constitute the ECM of human skin, mainly collagen type I and III, it can be easily sterilized and stored [16,18,35]. 

Lv K et al. [36] sought to compare the biodegradability and biocompatibility of Tilapia skin acellular dermal matrix to fetal bovine skin acellular dermal matrix through in vitro and in vivo implantation and cell culture techniques. The biocompatibility of Tilapia acellular dermal matrix was confirmed through in vitro and in vivo experiments. The Tilapia matrix scaffold was found to have no cytotoxicity and was friendly to cell growth, as shown by cell proliferation and cytotoxicity tests. L929 culture cells were able to infiltrate and adhere to the Tilapia acellular dermal matrix scaffold. In subcutaneous implantation in rats, the scaffold had tissue fluid infiltration and adhesion to the surrounding tissue within three days, and gradually degraded to form a tissue mass within 28 days, with no adverse reactions observed. Histological staining showed that the scaffold was integrated with the skin and cells like normal soft tissue, with the collagen fiber structure becoming less intact over time due to phagocytosis and degradation. 

One very important disadvantage of fish-derived collagen products is the lower denaturation temperature in comparison to mammalian counterparts. Although this threshold is indeed at room temperature for fish collagen, crosslinking during manufacturing processes could potentially increase it [37,38]. This characteristic is important both during manufacturing as well as for the clinical outcome.

Our study has shown that there is a significant difference between the collagen content of ON and OM skin. In addition, we observed that the sterilization process using sequentially increasing concentrations of glycerol interfered with the amount of collagen type I and III detected in the samples at the end of the said process. 

One interesting result is that collagen denaturation did not follow a linear pattern. Moreover, the mean area of collagen type I and III has increased with respect to the preceding ones. The concentration of Glycerol that remains trapped on the surface of the skin might have played a role in the diffraction of the polarized light. We searched for other possible explanations and have found that native collagen tends to bind glycerol to its surface, which stabilizes the triple helical structure [39,40]. Also, glycerol seems to have the role to sequestrate water [41,42], thus possibly interfering with the detection method. Further studies which incorporate immunohistochemical or biochemical determinations are needed. Furthermore, Lattouf et al. [43] questioned the capacity of Sirius red picrate staining under polarized light, to differentiate between type I and III collagen fibrils. In their study regarding Ehlers-Danlos type IV patients, Sirius red picrate managed to stain two types of collagen, corresponding to type I and III, but immunohistochemical staining did not show signs of type III collagen’s presence. This could’ve happened in part because although there was no collagen type III, the matrix presented aberrant pro-collagen type III which might have been stained by Sirius red. On the other hand, immunohistochemistry will not be able to detect collagen type III with its specific antibody, if there are only pro-collagen fibrils.

In our study, the ON subjects were immature specimens. One difference that we could see is that the ratio between type I and type III collagen was lower for the OM group, meaning that the amount of collagen type III was higher in young specimens, compared to the mature ones described by Alves et al. [16].

Normally, in the early stages of skin healing, collagen type III is more abundant than collagen type I [44,45]. This shows that young fish provide a closer match to the matrix physiologically produced in the early repair processes. Therefore, the use of skin harvested from young fish might be more helpful in the management of burn wounds. Further in vivo studies are recommended. 

### Limitations

Limits to our study are represented by the small sample size and the fact that our determination was done using histochemical quantitative methods and not biochemical or immuno-histochemical determinations. We could also prove that the sterilization method is effective against bacteria, but the determination of viral contents was not performed.

Further studies need to assess the exact amount of collagen within these specimens to have a more accurate image of their possible use in burn injuries.

## 5. Conclusions

The collagen content of OM skin was greater than that of ON in the natural state. The analysis of the sterilization process showed that the difference in collagen content between baseline (natur) and 99% Glycerol state was most probably caused by the interaction of glycerol with the collagen fibers, in the way that it changes their optical properties. Based on our results, OM skin harvested from young fish, with its greater collagen type I and III content per square µm, could potentially be a better candidate than ON skin, for use as a biologic skin scaffold in the treatment of burn wounds.

## Figures and Tables

**Figure 1 medicina-59-01002-f001:**
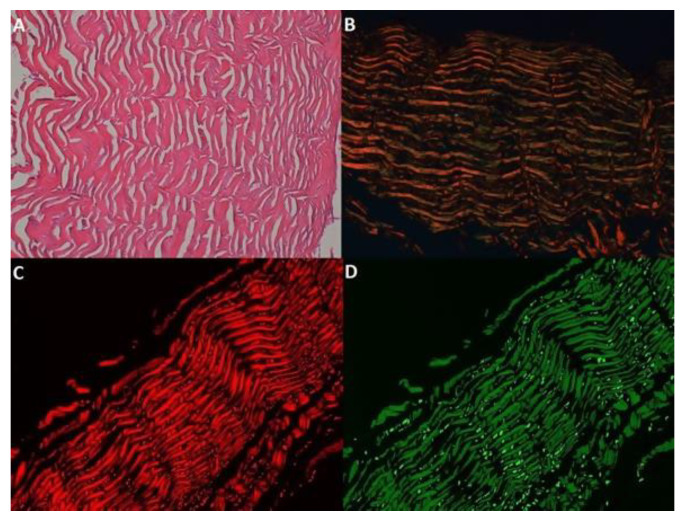
Multiple determination of the same skin sample. Photos represent the same sample in different histochemical coloring. (**A**) HE image. (**B**) Sirius red picrate image on polarized light microscopy. (**C**) RGB filtered image depicting collagen type I fibers. (**D**) RGB filtered image depicting collagen type III fibers.

**Figure 2 medicina-59-01002-f002:**
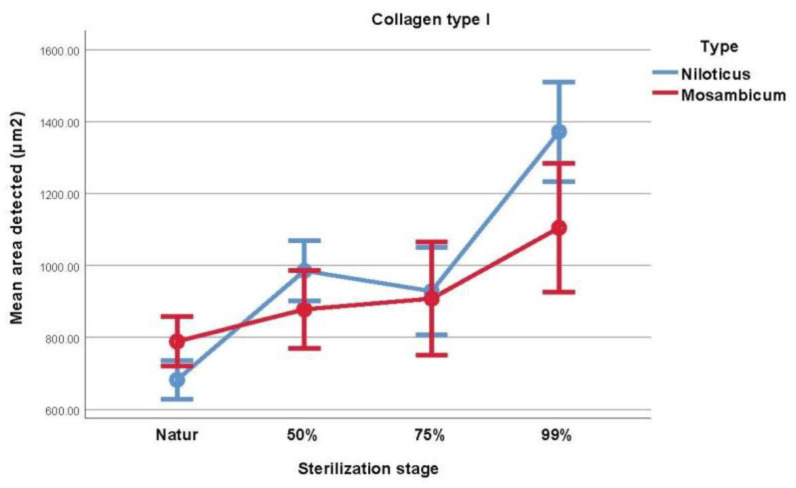
Visual comparison of the mean area of collagen type I from each type of fish.

**Figure 3 medicina-59-01002-f003:**
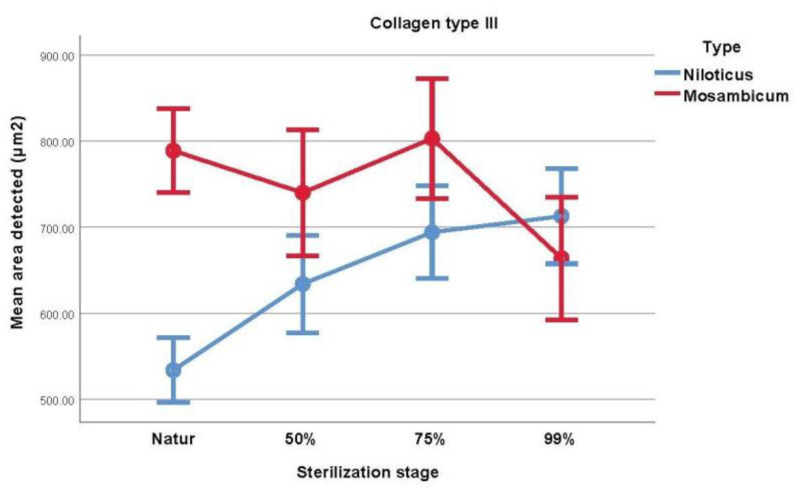
Visual comparison of the mean area of collagen type III from each type of fish.

**Figure 4 medicina-59-01002-f004:**
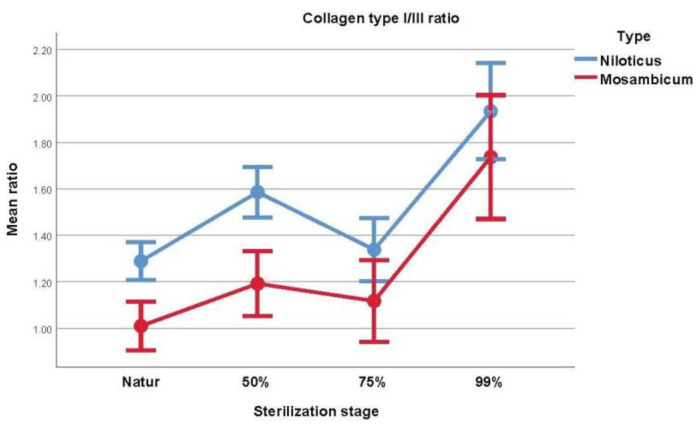
Visual comparison of the collagen type I/III ratio from each type of fish.

**Table 1 medicina-59-01002-t001:** Steps of the sterilization process.

Groups	Intervention	Time
1	Fat and muscle scrapping and 0.9% saline wash (Natural state)	N/A
2	2 × 2% Chlorhexidine baths + Solution: 50% Glycerol–49% Saline (0.9%) 1% PMF (Penicillin + Metronidazole + Fluconazole solution) +	2 × 30 min 24 h at 4 °C
3	Solution: 75% Glycerol–24% Saline (0.9%) 1% PMF +	3 h at 37 °C
4	Solution: 99% Glycerol–1% PMF	3 h at 37 °C

**Table 2 medicina-59-01002-t002:** Comparison of the mean area of collagen type I from each type of fish.

Sterilization	(I) Type	(J) Type	Mean Difference (I–J)	Std. Error	Sig.	95% Confidence Interval for Difference	
						Lower	Upper
Natur	Niloticus	Mosambicus	−106.683 *	43.154	0.017	−193.547	−19.818
50% Glycerol	Niloticus	Mosambicus	107.243	68.461	0.124	−30.562	245.048
75% Glycerol	Niloticus	Mosambicus	20.591	99.001	0.836	−178.687	219.869
99% Glycerol	Niloticus	Mosambicus	267.085 *	112.275	0.022	41.087	493.083

* The mean difference was significant at the 0.05 level. The unit of measure was µm^2^.

**Table 3 medicina-59-01002-t003:** Comparison of the mean area of collagen type I from each sterilization stage, within group.

Type	Sterilization Stage		Mean Difference	Std. Error	Sig.	95% Confidence Interval for Difference	
						Lower Bound	Upper Bound
Niloticus	Natur	50% Glycerol	−303.084 *	43.812	<0.001	−391.273	−214.894
		75% Glycerol	−246.303 *	63.687	<0.001	−374.498	−118.107
		99% Glycerol	−689.578 *	71.680	<0.001	−833.862	−545.294
	50% Glycerol	75% Glycerol	56.781	66.035	0.394	−76.141	189.703
		99% Glycerol	−386.494 *	77.439	<0.001	−542.371	−230.618
	75% Glycerol	99% Glycerol	−443.275 *	87.093	<0.001	−618.584	−267.967
Mossambicus	Natur	50% Glycerol	−89.158	56.562	0.122	−203.011	24.695
		75% Glycerol	−119.029	82.220	0.154	−284.529	46.471
		99% Glycerol	−315.810 *	92.538	0.001	−502.080	−129.540
	50% Glycerol	75% Glycerol	−29.871	85.251	0.728	−201.473	141.731
		99% Glycerol	−226.652 *	99.973	0.028	−427.887	−25.416
	75% Glycerol	99% Glycerol	−196.781	112.436	0.087	−423.103	29.542

* The mean difference was significant at the 0.05 level. The unit of measure was µm^2^.

**Table 4 medicina-59-01002-t004:** Comparison of the mean area of collagen type III from each type of fish.

Sterilization	(I) Type	(J) Type	Mean Difference (I–J)	Std. Error	Sig.	95% Confidence Interval for Difference	
						Lower	Upper
Natur	Niloticus	Mosambicus	−255.257 *	30.623	<0.001	−316.899	−193.616
50% Glycerol	Niloticus	Mosambicus	−106.039 *	45.942	0.026	−198.515	−13.563
75% Glycerol	Niloticus	Mosambicus	−109.035 *	43.777	0.016	−197.153	−20.917
99% Glycerol	Niloticus	Mosambicus	49.084	44.883	0.280	−41.261	139.428

* The mean difference was significant at the 0.05 level. The unit of measure was µm^2^.

**Table 5 medicina-59-01002-t005:** Comparison of the mean area of collagen type III from each sterilization stage, within group.

Type	Sterilization Stage		Mean Difference	Std. Error	Sig.	95% Confidence Interval for Difference	
						Lower Bound	Upper Bound
Niloticus	Natur	50% Glycerol	−100.379 *	28.407	0.001	−157.559	−43.199
		75% Glycerol	−160.451 *	34.668	<0.001	−230.233	−90.669
		99% Glycerol	−179.153 *	33.035	<0.001	−245.648	−112.657
	50% Glycerol	75% Glycerol	−60.072	31.775	0.065	−124.032	3.887
		99% Glycerol	−78.774 *	27.496	0.006	−134.121	−23.427
	75% Glycerol	99% Glycerol	−18.702	23.347	0.427	−65.697	28.293
Mossambicus	Natur	50% Glycerol	48.840	36.673	0.190	−24.979	122.659
		75% Glycerol	−14.229	44.756	0.752	−104.317	75.860
		99% Glycerol	125.188 *	42.648	0.005	39.343	211.033
	50% Glycerol	75% Glycerol	−63.069	41.021	0.131	−145.640	19.503
		99% Glycerol	76.348 *	35.497	0.037	4.896	147.801
	75% Glycerol	99% Glycerol	139.417*	30.141	<0.001	78.747	200.087

* The mean difference was significant at the 0.05 level. The unit of measure was µm^2^.

**Table 6 medicina-59-01002-t006:** Comparison of the collagen type I/III ratio from each type of fish.

Sterilization	(I) Type	(J) Type	Mean Difference (I–J)	Std. Error	Sig.	95% Confidence Interval for Difference	
						Lower	Upper
Natur	Niloticus	Mosambicus	0.280 *	0.066	<0.001	0.147	0.412
50% Glycerol	Niloticus	Mosambicus	0.394 *	0.088	<0.001	0.217	0.571
75% Glycerol	Niloticus	Mosambicus	0.221	0.110	0.050	0.000	0.443
99% Glycerol	Niloticus	Mosambicus	0.198	0.168	0.244	−0.140	0.535

* The mean difference was significant at the 0.05 level. The unit of measure was µm^2^.

**Table 7 medicina-59-01002-t007:** Comparison of the mean area of collagen type III from each sterilization stage, within group.

Type	Sterilization Stage		Mean Difference	Std. Error	Sig.	95% Confidence Interval for Difference	
						Lower Bound	Upper Bound
Niloticus	Natur	50% Glycerol	−0.297 *	0.072	<0.001	−0.441	−0.153
		75% Glycerol	−0.049	0.069	0.480	−0.189	0.090
		99% Glycerol	−0.646 *	0.115	<0.001	−0.877	−0.415
	50% Glycerol	75% Glycerol	0.248 *	0.087	0.006	0.073	0.422
		99% Glycerol	−0.349 *	0.112	0.003	−0.575	−0.123
	75% Glycerol	99% Glycerol	−0.597 *	0.122	<0.001	−0.842	−0.351
Mossambicus	Natur	50% Glycerol	−0.182	0.092	0.054	−0.368	0.003
		75% Glycerol	−0.108	0.089	0.234	−0.288	0.072
		99% Glycerol	−0.728 *	0.148	<0.001	−1.026	−0.429
	50% Glycerol	75% Glycerol	0.075	0.112	0.508	−0.151	0.300
		99% Glycerol	−0.545 *	0.145	<0.001	−0.837	−0.253
	75% Glycerol	99% Glycerol	−0.620 *	0.157	<0.001	−0.937	−0.303

* The mean difference was significant at the 0.05 level. The unit of measure was µm^2^.

## Data Availability

Data available on request from the corresponding author.

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
