# Peer review of "Comparison of Type I and Type III Collagen Concentration between Oreochromis mossambicus and Oreochromis niloticus in Relation to Skin Scaffolding"

_medicina, 2023, doi:10.3390/medicina59061002_

Round 1

Reviewer 1 Report

The article is written with good grammar. Limitations are stated in the article. The difference between both fish breeds was clearly demonstrated in the study. The collagen content of OM skin has proven to be more than ON one. The role of collagen 1 and 3 in wound healing is known. It was nice to present the data with histochemical analysis and graphics in the study. Many thanks for your great effort.

The main question addressed by the research is based on examining the difference in collagen concentrations of the samples obtained from  Nile Tilapia (Oreochromis niloticus)(ON) and Tilapia Mossambica (Oreochromis mossambicus) (OM) fish for skin scaffolding.

Although the skin of ON has been suggested in the literature as a possible biological scaffold for use in burns, it was aimed to examine the OM type, which does not have much data in the literature, and to compare ON and collagen types I and III content. In addition, the differences between collagen type I and collagen type III ratios in both types have been recently brought to the literature by the research of this article. Congratulations to the authors.

As the authors noted in the limitation section, a bigger sample size could be used in the study. Biochemical and immunohistochemical analyzes could be used.

The figures and tables are appropriate for the methodology and results. 

Author Response

Thank you for your suggestions and comments. A word document has been attached.

Reviewer 2 Report

Dear authors, 

I have studied with great interest the manuscript “Collagen concentration comparison between Oreochromis mossambicus and Oreochromis niloticus in regards to skin scaffolding”. This study compares the type I and type III collagen content of the skin of two species of fish of the Oreochromis family.

I have some comments: 

I would recommend modifying the title to ''Comparison of type I and type III collagen concentration between Oreochromis mossambicus and Oreochromis niloticus in relation to skin scaffolding.''

-The determination of collagen concentration was performed by quantitative histochemical methods. It would be necessary, as the authors mention in the limitations of their study, to perform biochemical and immunohistochemical determinations. It would also be interesting to perform genetic analysis (PMID: 32761644). Histochemical analysis is a good method of characterization, although it must be complementary to many others in order to have a deep and meaningful investigation.

-Another major limitation of the study is the small sample size (n=10), which in the end was n=8, since two were lost. Have any analyses been performed to calculate the significant sample size? If so, it should be indicated in material and methods.

The study is a good approximation, however, this experiment should be a small part of a larger group of experiment that leads to reliable conclusions.

Author Response

(The authors gave the same response as above.)

Reviewer 3 Report

Dear Author,

Article entitled Collagen concentration comparison between Oreochromis mos- 2 sambicus and Oreochromis niloticus in regards to skin scaf- 3 folding is of interest of researcher 

There are some recommendation need to be consider

1. Refer line number 28,29 and 79,80. Data mismatch

2. Study design/ methodology is need to describe clearly

3. Scientific writing need to improve

4. Section 2.3 Method and results part need to be at appropriate section.

5. Justify basis of selection of number of fish/ Sample size

6. Section 2.2 No any citation mentioned for methodology

7. Lot of typo errors

8. Figure qualities need to be improve

Minor editing of English language required

Author Response

(The authors gave the same response as above.)

Round 2

Reviewer 2 Report

Thank you for your reply, the manuscript is suitable for publication.

Author Response

We, the authors thank you very much for your review and for accepting our work for publishing.

Reviewer 3 Report

1. Refer line number 28,29 and 79,80. Data mismatch: Revision Accepted

2. Study design/ methodology is need to describe clearly: Revision Accepted

3. Scientific writing need to improve: Revision Accepted

4. Section 2.3 Method and results part need to be at appropriate section: Partial corrections done. 2.3 Figure and results data appearing under material section.

5. Justify basis of selection of number of fish/ Sample size: Need Clarity in sample size (Numbers)/ Any Guidelines followed?

6. Section 2.2 No any citation mentioned for methodology: Citation needed

7. Lot of typo errors: Partially Corrected

8. Figure qualities need to be improve: Figure Clarity /Quality can be improved

Minor editing of English language/Typo errors corrections required

Author Response

Thank you, please find the point by point responses attached.
